# Differential Effects of Anti-TNFα and Anti-α4β7 Drugs on Circulating Dendritic Cells Migratory Capacity in Inflammatory Bowel Disease

**DOI:** 10.3390/biomedicines10081885

**Published:** 2022-08-04

**Authors:** Irene Soleto, Samuel Fernández-Tomé, Irene Mora-Gutiérrez, Montserrat Baldan-Martin, Cristina Ramírez, Cecilio Santander, José Andrés Moreno-Monteagudo, María José Casanova, Fernando Casals, Sergio Casabona, Irene Becerro, María Chaparro, David Bernardo, Javier P. Gisbert

**Affiliations:** 1Gastroenterology Unit, Centro de Investigación Biomédica en Red de Enfermedades Hepáticas y Digestivas (CIBERehd), Hospital Universitario de La Princesa, Instituto de Investigación Sanitaria Princesa (IIS-Princesa), Universidad Autónoma de Madrid (UAM), 28006 Madrid, Spain; 2Departamento de Nutrición y Ciencia de los Alimentos, Facultad de Farmacia, Universidad Complutense de Madrid, 28040 Madrid, Spain; 3Biobanco del Sistema de Salud de Aragón, 50009 Zaragoza, Spain; 4Mucosal Immunology Lab, Centro de Investigaciones Biomédicas en Red de Enfermedades Infecciosas (CIBERINFEC), Unidad de Excelencia Instituto de Biología y Genética Molecular (IBGM), Universidad de Valladolid, 47005 Valladolid, Spain; 5Centro de Investigación Biomédica en Red de Enfermedadmes Infecciosas (CIBERINFEC), 28029 Madrid, Spain

**Keywords:** biological drugs, inflammatory bowel disease, dendritic cells, migration, intestinal mucosa, anti-TNFα, vedolizumab

## Abstract

Inflammatory bowel disease (IBD) is an idiopathic and chronic disorder that includes ulcerative colitis (UC) and Crohn’s disease (CD). Both diseases show an uncontrolled intestinal immune response that generates tissue inflammation. Dendritic cells (DCs) are antigen-presenting cells that play a key role in tolerance maintenance in the gastrointestinal mucosa. Although it has been reported that DC recruitment by the intestinal mucosa is more prominent in IBD patients, the specific mechanisms governing this migration are currently unknown. In this study, the expression of several homing markers and the migratory profile of circulating DC subsets towards intestinal chemo-attractants were evaluated and the effect of biological drugs with different mechanisms of action, such as anti-TNFα or anti-integrin α4β7 (vedolizumab), on this mechanism in healthy controls (HCs) and IBD patients was also assessed. Our results revealed that type 2 conventional DCs (cDC2) express differential homing marker profiles in UC and CD patients compared to HCs. Indeed, integrin β7 was differentially modulated by vedolizumab in CD and UC. Additionally, although CCL2 displayed a chemo-attractant effect over cDC2, while biological therapies did not modulate the expression of the homing markers, we paradoxically found that anti-TNF-treated cDC2 increased their migratory capacity towards CCL2 in HCs and IBD. Our results therefore suggest a key role for cDC2 migration towards the intestinal mucosa in IBD, something that could be explored in order to develop novel diagnostic biomarkers or to unravel new immunomodulatory targets in IBD.

## 1. Introduction

Inflammatory bowel disease (IBD) is an idiopathic and chronic disorder that includes ulcerative colitis (UC) and Crohn’s disease (CD). Although both diseases differ in their location, type of inflammation and symptoms, they develop as the consequence of a pathological response to both the innate and the adaptive immune systems against harmless antigens, including commensal microbiota and food antigens, generating an inflammation of the gastrointestinal (GI) tract. Currently, IBD is a global disease affecting more than 2 million people in the United States and more than 3.2 million people in Europe [1]. Indeed, some studies have suggested that it may affect 1 out of 125 individuals in western countries, and its incidence is on the rise [2,3,4]. Unfortunately, there is no cure for CD or UC, so the main therapeutic objective is to induce clinical remission of the disease.

Biological treatments, which are considered the most potent drugs for IBD treatment, are efficient at inducing remission in only one-third of patients. Indeed, another one-third of patients experience clinical improvement but without fully reaching control of the inflammatory process, while the last third of patients do not respond at all to treatment [5]. Furthermore, biological drugs are expensive. Indeed, within Europe, it is estimated that the cost to health care is EUR 4.6–5.6 billion per year [6]. If the medical treatment fails, patients must usually undergo surgery. A high proportion of patients also have recurrent disease, which often leads to repeated bowel resections, which consequently leads to the risk of short bowel syndrome, increased disability, a negative social impact and increased healthcare costs in these patients.

In recent years, the understanding of the role of some cytokines involved in IBD pathogenesis has deepened, facilitating the development of biological drugs against specific immune molecules. Biological treatments approved for both CD and UC act against the tumor necrosis factor TNF-α (infliximab, adalimumab, certolizumab pegol and golimumab), as well as against α4β7 integrin (vedolizumab) and the p40 subunit of the interleukins IL-12 and IL-23 (ustekimumab) [7,8]. Moreover, new small molecules have been developed, such us tofacitinib, which is a pan-inhibitor of the janus kinase (JAK) pathway, although the latter has been only approved in UC patients [9]. However, the exact mechanisms of action of these biological treatments are still unknown. Although these compounds usually modulate the tissue cytokine milieu and, therefore, the phenotype and function of circulant and tissue resident immune cells [10,11,12], others, such as vedolizumab (monoclonal antibody against α4β7 integrin), prevent the immune cell migration towards the GI mucosa [13]. Nevertheless, it is currently unknown whether anti-inflammatory biological drugs can also modulate this migration mechanism.

In this regard, dendritic cells (DCs) are phagocytic professional antigen-presenting cells that link the innate and the adaptive immune systems. Indeed, they are a unique cell type able to stimulate naïve T cells into generating an antigen-specific immune response. In humans, DCs can be divided into plasmacytoid dendritic cells (pDCs: CD123^+^ BDCA2^+^) and classical or conventional dendritic cells (cDCs: CD11c^+^), the latter being further subdivided into type 1 (cDC1: CD141^+^ CXCR1^+^), that is specialized in cross presentation and type 2 (cDC2: CD1c^+^ SIRPα^+^), that is involved in classical antigen presentation [14,15]. In the GI tract, cDCs maintain mechanisms of immune tolerance towards nutrients and commensals [16,17]. However, the process is dysregulated in IBD where they promote the development of pro-inflammatory T cells. The alteration in human intestinal DCs during UC enhances Th2 immunity, rendering a loss of cytokine production involved in epithelial barrier maintenance [18]. Indeed, cDCs are thought to migrate to GI tissue through the expression of the surface integrin α4β7, which interacts with Mucosal Vascular Addressin Cell Adhesion Molecule 1 (MadCam1), expressed by vascular endothelial cells. Furthermore, they also migrate by CCR9 binding to its ligand (CCL25), expressed by small bowel epithelial cells, as well as by CCR2 [19]. Therefore, although the cDC migration towards the GI mucosa is enhanced in IBD [20,21], (with the cDC homing marker expression correlating with the phenotype of the disease [22]) the mechanisms underlying these migrations and whether biological drugs can modulate them are currently unknown. Therefore, this study aimed to analyze the cDC and pDC migratory capacity towards the GI mucosa and whether biological drugs can modulate this mechanism in healthy controls and IBD.

## 2. Materials and Methods

### 2.1. Patients and Sample Collection

Biological samples were obtained from a total of 72 individuals, including 15 healthy controls (HCs), 13 patients with UC with endoscopic inflammation (active, aUC), 15 patients with UC without endoscopic inflammation (quiescent, qUC), 15 patients with CD with endoscopic inflammation (active, aCD) and 14 patients with CD without endoscopic inflammation (quiescent, qCD). HCs were patients referred due to changes in bowel transit, colorectal cancer screening or rectal bleeding. Nevertheless, they all had macroscopically and histologically normal mucosa and lacked known inflammatory, autoimmune or malignancy diseases. The demographic data of the patients can be found in Appendix A. The study was approved by the local ethics committee at La Princesa Hospital (Madrid, Spain). All patients gave written informed consent for sample collection. From each individual, 20 mL of blood was obtained and immediately processed in the laboratory.

### 2.2. Blood Processing

Peripheral blood mononuclear cells (PBMCs) were obtained by centrifugation using Ficoll–Paque PLUS (Amersham Biosciences, Buckinghamshire, UK). The PBMCs were washed twice in a complete medium (RPMI 1640 (Sigma-Aldrich, Burlington, MA, USA) consisting of 100 μg/mL penicillin/streptomycin, 2 mM L-glutamine, 50 ug/mL gentamicin (Sigma-Aldrich) and 10% fetal bovine serum (TCS cellworks, Northampton, UK)). Then, the PBMCs were stained in PBS containing 1mM EDTA and 0.02% sodium azide (FACS buffer) with fluorochrome-conjugated antibodies, as explained below.

### 2.3. Antibody Labelling

The PBMCs were stained with monoclonal antibodies and characterized by flow cytometry. In all cases, a Live/Dead fixable near-IR dead cell stain kit (Molecular Probes, Eugene, OR, USA) was added to the cells for 1 min at room temperature prior to performing the antibody staining, hence allowing the exclusion of dead cells from the analysis. Appendix A shows the specificity, clone, fluorochrome and sources of the antibodies used. Cells were labeled in a FACS buffer on ice and in the dark for 20 min following a nonspecific binding block. The circulating pDCs, cDC1 and cDC2 were identified within singlet viable leukocytes and further assessed for the expression of different homing markers (CCR2, CCR5, CCR6, CCR9 and β7) related to the migration towards the GI tract.

### 2.4. PBMC Culture

The PBMCs from HCs and IBD patients were also cultured in a complete medium (1 million PBMCs/1 mL) in a 24-well cell culture plate at 37 °C for 18 h in the presence of an anti-TNF drug (infliximab, inflectra and adalimumab, 10 μg/mL) and anti-α4β7 drug mAb (vedolizumab, 100 μg/mL). As an internal and negative control, the paired PBMCs were cultured in a complete medium in the absence of any drug.

### 2.5. Transwell Migration Experiments

The migratory capacity of circulating DC subsets from the different study groups (HCs/IBD; with/without conditioning) was determined using 3 μm pore polycarbonate membrane culture inserts (transwell). The migration towards the complete culture medium (basal control) or medium supplemented with 100 ng/mL CCL2 (CCR2 ligand), 500 ng/mL CCL25 (CCR9 ligand) or 1 μg/mL MadCAM1 (β7 ligand) was assessed. Briefly, cells from different groups with or without conditioning were seeded in the apical chamber at 200.000 PBMCs per well. Then, in the basal chamber, 200 µL of the medium with the corresponding chemo-attractant was added. After 4 h, the migrated cells in the basal chamber were further harvested and analyzed by flow cytometry.

### 2.6. Flow Cytometry and Data Analysis

The cells were analyzed using an LSR–Fortessa (BD Biosciences, San Jose, CA, USA) for the DC characterization and on a BD Canto II flow cytometer (BD Biosciences) for the migration assays. In all cases, the results were analyzed using FlowJo version 10.1 (Becton Dickenson and Company, Ashland, OR, USA). All cells were analyzed within the singlet viable fraction. Positive and negative gates were set by the FMO method.

### 2.7. Statistical Analysis

The data were analyzed using GraphPad Prism 6.01 software (San Diego, CA, USA) by one-way or two-way analysis of variance (ANOVA) (with or without repeated measures). The subsequent post hoc correction for multiple comparisons was applied when required, as detailed in each figure legend. The significance threshold was fixed at *p* < 0.05 in all cases.

## 3. Results

### 3.1. Differential Migration Pattern in Dendritic Cell Subsets from HCs and IBD Patients

Human circulating DC subsets were identified within singlet viable CD19^−^HLA-DR^+^CD14^−^ cells. The pDCs were identified as CD123^+^ while cDCs were identified as CD11c^+^. The latter were further subdivided into cDC1 (CD141^+^) and cDC2 (CD1c^+^), based on the expressions of CD141 and CD1c, respectively (Figure 1A). All DC subsets were further characterized for the expressions of β7, CCR2, CCR5, CCR6 and CCR9, revealing that cDC2 expressed higher levels of β7, CCR5 and CCR6, compared to pDCs and cDC1 (Figure 1B). Furthermore, cDC1 showed a lower expression of CCR2, compared to its pDC and cDC2 counterparts.

Having characterized the resting levels of these homing markers in healthy controls, we studied whether their expression was altered in IBD patients including aUC, qUC, aCD and qCD. First, we found that the homing profile of pDCs did not change among the different study groups. Moreover, the percentage of integrin β7^+^ cells was not altered in any DC subset either. On the contrary, cDC2 from aUC patients showed a lower expression of both CCR2 and CCR6 compared to HCs and aCD, while the expression of CCR9 within this subset was expanded in both quiescent groups (qUC and qCD) compared to HCs and their respective inflamed counterparts. Lastly, CCR5 and CCR6 were decreased on cDC1 in aUC patients compared to HCs and qUC (CCR5) and to qUC (CCR6) (Figure 2).

### 3.2. DC Subset Migration towards GI Chemo-Attractants

Having described the migratory profile of DC subsets in HCs and IBD, we focused on the expressions of CCR2, CCR9 and integrin β7 as the most relevant markers related to the DC subset migration towards the GI tract. Therefore, in order to assess whether their expression was functional, we performed transwell migration assays towards their respective ligands (CCL2, CCL25 and Madcam1, respectively). Results for each subset in each study group were relativized with respect to the spontaneous migration towards a non-supplemented culture medium denoted as the basal migratory capacity. The number of different DC subsets migrated are indicated in Appendix A.

Only CCL2 showed a statistically significant chemo-attractant capacity (referred to as the basal migratory capacity, data not shown) over pDCs and cDC2, not just in HCs but also in IBD (Figure 3). Indeed, the increased migration of cDC2 towards CCL2 was observed in all IBD groups (CD/UC; active/quiescent), while in pDCs, this increase was only observed in patients with qUC and aCD. CCL25 and MadCam1, on the contrary, did not exhibit any statistically significant chemo-attractant capacity over any circulating DC subset from HCs or IBD patients.

### 3.3. Vedolizumab but Not Anti-TNF Drugs Modify the Surface Expression of Migratory Markers on DCs

Taking into account that the different DC subsets from IBD patients showed an altered expression pattern of migratory receptors and a different migratory capacity towards GI chemo-attractants, we studied whether this mechanism could also be modulated with biological drugs. Hence, the PBMCs from HCs and IBD patients were conditioned with three different anti-TNF drugs (infliximab, inflectra and adalimumab) and one anti-α4β7 (vedolizumab). None of the anti-TNF treatments modulated the expression of any of the studied homing markers in the different DC subsets (Appendix A). Therefore, for the subsequent studies, all three studied anti-TNF drugs were considered as a single treatment. On the contrary and as expected, vedolizumab showed a strong effect in reducing the expression of integrin β7 in all DC subsets in all groups, except for qCD and aUC, in both cDC1 and cDC2 (Figure 4).

### 3.4. Biological Drugs Modify the Migratory Capacity of Different Circulating DC Subsets

Finally, although biological treatments (except for vedolizumab) did not alter DC homing markers, we studied whether they could modulate the DC subset migration towards CCL2, CCL25 and MadCam1.

As above, pDCs only displayed a migration capacity towards CCL2, except in the case of patients with aUC. The pDC from HCs conditioned with vedolizumab showed a higher migratory capacity towards CCL2 than cells incubated with an anti-TNF. In a similar manner, pDCs from qUC patients exhibited a higher migratory capacity compared with the basal condition (untreated cells towards a non-supplemented culture medium) in all cases. The pDCs from CD patients, both aCD and qCD, showed a higher migratory capacity when cultured in the presence of an anti-TNF than in the basal condition. In the case of aCD patients, this effect was also shown in cells cultured with vedolizumab.

When we focused on cDC2, we found that conditioning with an anti-TNF increased their capacity to migrate towards CCL2 in all study groups except for qCD. Moreover, an anti-TNF-treated cDC2 also displayed an increased migration towards CCL25 and MadCam1 in patients with aUC, revealing an increased paradoxical migratory capacity of these cells in the presence of an anti-TNF. Lastly, the cDC1 migration capacity was not modulated by any of the studied biological treatments (Figure 5).

## 4. Discussion

We hereby describe how the anti-α4β7 drug modulates the DC migratory capacity towards CCL2 in IBD and HCs, thereby expanding our current knowledge on the mechanisms of action of biological drugs, as well as on the surface markers implicated in the recruitment of circulating immune cells by the GI mucosa, thereby unveiling future targets for new treatments for IBD.

Our results showed that the percentage of cDC2 that expresses homing markers is higher than that of other DC subsets, indicating that these markers can be used as targets to rationally design new drugs in order to prevent the recruitment of these cells to the GI mucosa. Indeed, Canavan et al. studied the essential role of cDC2 in inflammatory arthritis and established the need to develop therapeutic interventions focusing on these cells [23]. In agreement with our results, Bernardo et al. demonstrated that CCR2 mediates the cDC2 migration to the colonic mucosa [19], but paradoxically, the proportion of cDC2 from aUC patients expressing CCR2 and CCR6 is lower than that from HCs and aCD patients. On the contrary, the proportion of cDC2 expressing CCR9 is higher both in patients with qUC or qCD. A CCR9 protective role against IBD was also described in a previous study from our group [24]. Therefore, the expression of these markers on cDC2 should be evaluated as a potential biomarker in order to discriminate between patients with a quiescent or active disease, which is potentially useful in order to avoid an unnecessary colonoscopy and instead using multiparameter flow cytometry, which is a well-established technique in the diagnosis of other diseases, such as B cells malignancies [25]. In this regard, it has been published with regards to murine models, that CCR9^−/−^ mice are more susceptible to DSS-induced colitis [26]. In addition, a clinical trial of the anti-MadCam1 drug revealed that CCR9 was a relevant pharmacodynamic biomarker as it was increased following treatment in aCD patients [27]. In an attempt to evaluate this hypothesis in our study, a correlation analysis between the percentage of cDC2 expressing CCR9 and the endoscopic index of IBD patients was carried out with negative results (data not shown), probably due to the higher inter-individual variability and the number of patients per group; accordingly, a deeper study is needed in order to properly assess this hypothesis. Interestingly, integrin β7 was the only homing marker that remained unmodulated in all DC subsets in IBD compared with HCs. Although vedolizumab exerts its benefits by blocking this molecule, other biological drugs, such as etrolizumab, did not demonstrate any benefit compared with a placebo in the EUCALPTUS clinical trial; it may be because of that, that endothelial targets are being evaluated in order to develop new pharmacological tools [28].

Taking into account that the migration markers of DC subsets differ among the studied patient groups, we also analyzed the migratory capacity of these subsets towards GI chemo-attractants. What we saw is that only CCL2 was capable of recruiting pDCs and cDC2 in both CD and UC patients. Maybe the effect of both CCL25 and MadCam1 is less evident because CCR2 is expressed in almost 100% of the DC subsets, while the CCR9 expression is residual and integrin β7 is only expressed in around 50% of the cells. In addition, CCR2 has been stipulated as an important homing cDC target in the colonic mucosa [19]. In this regard, we hereby have observed that CCL2 (one of the CCR2 ligands) exerts a high cDC chemo-attractant effect. Therefore, CCR2 could be evaluated as a new immunomodulatory target for UC and colonic CD. Furthermore, some in vivo works demonstrate a beneficial effect in experimental colitis when the number of CCR2^+^ cells is reduced in the inflamed colon and peripheral blood [29].

According to the effect of biological drugs on the expression of migratory markers on DC subsets, only integrin β7 was downregulated following treatment with vedolizumab, as expected, in all patients except those with qCD. Although it has been described that the MadCam1 expression is higher in patients with active IBD [30], this is the first time, to our knowledge, that a different behavior of integrin β7 expression is described between CD and UC patients. It has been previously reported that the integrin α4β1/VICAM-1 axis plays a dominant role during immune cell migration to the small intestine [31]. Therefore, the differential regulation of integrin β7 in CD patients may explain the differences in leukocyte homing between CD and UC patients.

Another major finding of our study was that cDC2 treated with an anti-TNF increased their migration capacity towards CCL2, suggesting that an anti-TNF treatment may increase the cDC2 migration capacity towards the GI mucosa [19]. Therefore, future studies should assess the maturation status of these cells as they may also display a regulatory profile by diminishing the mucosal inflammation, thereby providing a novel mechanism of action for these biological cells. In this regard, it has been published that the anti-TNF therapeutic response in UC patients is associated with reduced monocyte activation and CCL2 serum levels [32]. Accordingly, an anti-TNF treatment presents different mechanisms of action in order to diminish inflammation in the mucosa, but these mechanisms are not well understood [33,34,35]; hence, future studies should be carried out in order to try to describe all of these mechanisms.

In summary, the current study found that cDC2 could be considered as a potential diagnostic marker, given the differential expression of their homing markers between HCs and IBD. Indeed, the CCR9 expression on cDC2 should be evaluated in the future in order to corroborate its utility as a diagnostic marker between active and quiescent patients that could contribute to avoiding an unnecessary colonoscopy. Additionally, only integrin β7 is modulated by vedolizumab but not by any anti-TNFα in all groups except for CD patients, thereby demonstrating that the mechanism of immune cell recruitment in CD and UC is different. In addition, CCL2 showed a predominant role in DC recruitment being modulated by an anti-TNF but not by vedolizumab. However, the results derived from the current work should be validated by in vivo studies, to further understand the biological drugs’ mechanism of action and to elucidate future diagnostic and prognostic biomarkers that could then be implemented into clinical practice.

## Figures and Tables

**Figure 1 biomedicines-10-01885-f001:**
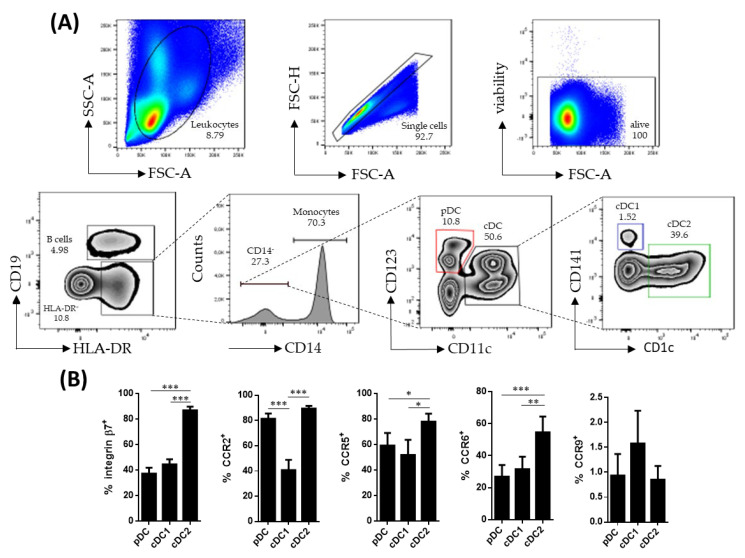
Characterization of circulating dendritic cell subsets. (**A**) Dendritic cell (DC) subsets were identified by flow cytometry within singlet viable peripheral blood mononuclear cells. HLA-DR^+^CD19^−^CD14^−^ cells were further subdivided into plasmacytoid dendritic cells (pDCs) (CD123^+^ CD11c^−^) and conventional dendritic cells (cDCs) (CD11c^+^). The cDC cells were further divided based on the expressions of CD141 (cDC1) and CD1c (cDC2). (**B**) Phenotype of the different DC subsets from healthy controls was determined by analyzing the expressions of integrin β7, CCR2, CCR5, CCR6 and CCR9. Results are expressed as the percentage of positive cells (%) (mean ± SEM *n* = 13–15) referred to as a fluorescence minus one. One-way ANOVA repeated measures with the Tukey correction was applied to compare the basal expression of the different markers between pDCs, cDC1 and cDC2. *p*-values < 0.05 were considered significant (* < 0.05, ** < 0.01, *** < 0.001).

**Figure 2 biomedicines-10-01885-f002:**
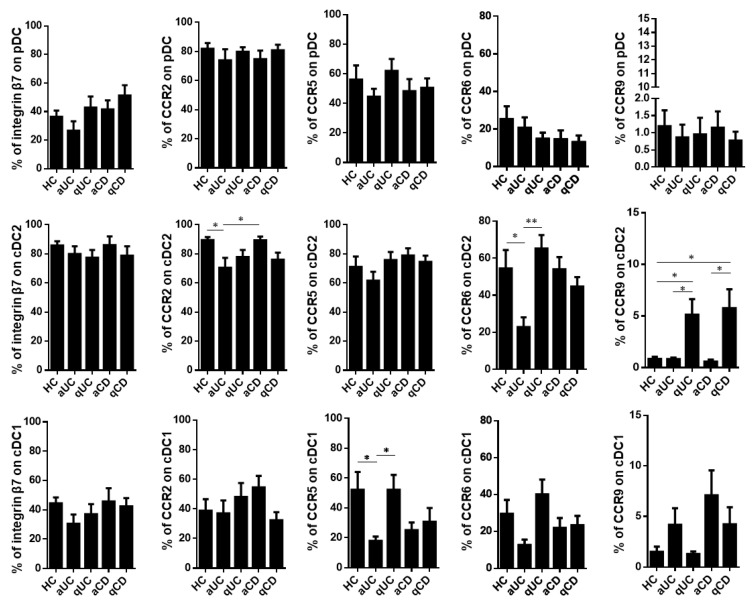
Integrin β7, CCR2, CCR5, CCR6 and CCR9 expressions on circulating dendritic cells from healthy controls and patients with inflammatory bowel disease. Dendritic cell (DC) subsets were identified as in Figure 1 in healthy controls (HCs) and in patients with active ulcerative colitis (aUC), quiescent ulcerative colitis (qUC), active Crohn’s disease (aCD) and quiescent Crohn’s disease (qCD). The expressions of β7, CCR2, CCR5, CCR6 and CCR9 were further determined in each subset within each study group. Results are shown as the percentage of positive cells (%) (mean ± SEM *n* = 13–15). One-way ANOVA with the Tukey correction was applied to compare integrin β7, CCR2, CCR5, CCR6 and CCR9 expression levels on pDCs, cDC2 and cDC1 between HCs and the different groups of inflammatory bowel disease patients. Furthermore, homing marker expressions were compared between quiescent and active patients for each disease in order to evaluate changes associated with the different mucosal statuses within the same disease. In addition, aUC and aCD, as well as qUC and qCD were compared in order to evaluate changes associated to different diseases with the same mucosal status. *p*-values < 0.05 were considered significant (* < 0.05, ** < 0.01).

**Figure 3 biomedicines-10-01885-f003:**
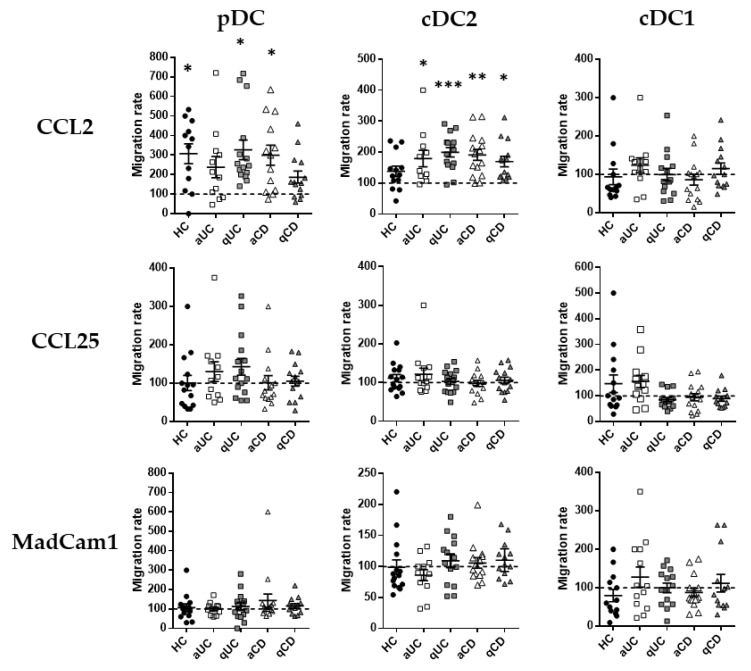
Dendritic cell subset migration towards CCL2 is increased in patients with inflammatory bowel disease. Peripheral blood mononuclear cells from healthy controls (HCs) and patients with active ulcerative colitis (aUC), quiescent ulcerative colitis (qUC), active Crohn’s disease (aCD) and quiescent Crohn’s disease (qCD) were allowed to migrate towards gut-homing chemo-attractants, including CCL2, CCL25 and MadCam1. Numbers of migrated DC subsets (pDCs, cDC2 and cDC1), as identified in Figure 1, were further determined. All results were relativized with respect to the spontaneous migration of the cells towards a non-supplemented culture medium (basal condition, dotted line) (mean ± SEM *n* = 15). One-way ANOVA with the Tukey correction was applied to compare with the basal condition. *p*-values < 0.05 were considered significant (* < 0.05, ** < 0.01, *** < 0.001).

**Figure 4 biomedicines-10-01885-f004:**
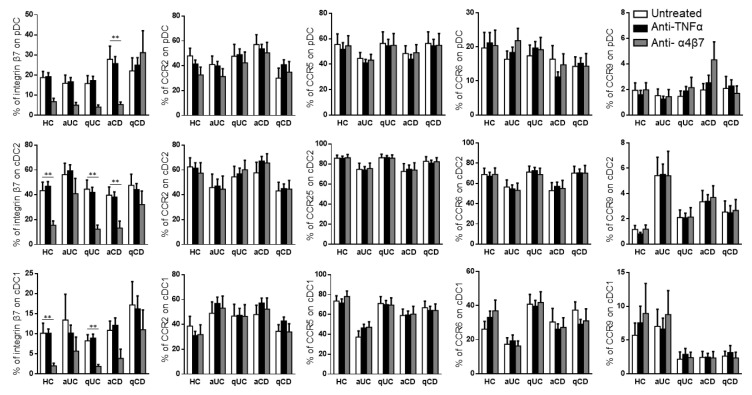
Vedolizumab downregulates the integrin β7 expression on circulating dendritic cell subsets. Dendritic cell (DC) subsets were identified as in Figure 1 and studied for the expressions of β7, CCR2, CCR5, CCR6 and CCR9 in healthy controls (HCs) and in patients with active ulcerative colitis (aUC), quiescent ulcerative colitis (qUC), active Crohn’s disease (aCD) and quiescent Crohn’s disease (qCD) following conditioning with an anti-TNF biological drug (black bars) or anti-α4β7 (grey bars). Results were compared with untreated controls (white bars). The percentages of positive cells were determined within each different subset (mean ± SEM *n* = 13–15). Two-way ANOVA with the subsequent post hoc correction was performed in order to compare integrin β7, CCR2, CCR5, CCR6 and CCR9 expression levels on pDCs, cDC2 and cDC1 from HCs and patients with a different disease status between cells subjected to different treatments, as described above. *p*-values < 0.05 were considered significant (** < 0.01).

**Figure 5 biomedicines-10-01885-f005:**
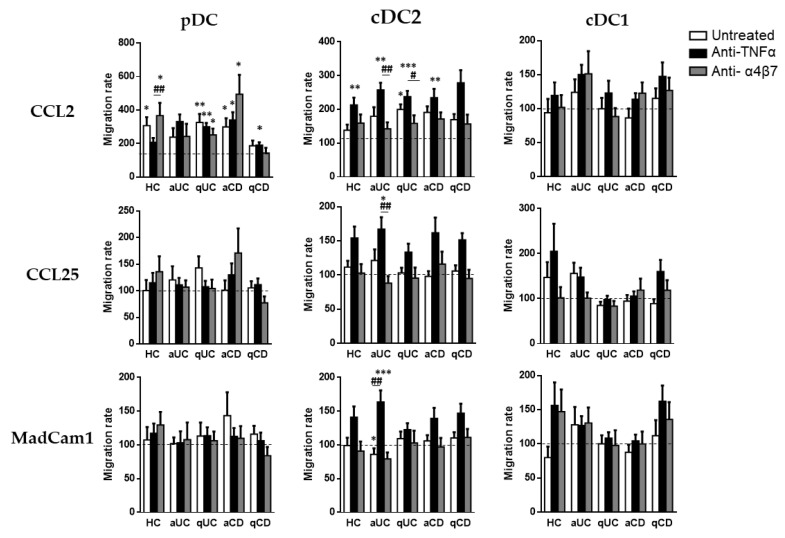
Biological treatments modulate the dendritic cell subset migratory capacity. Peripheral blood mononuclear cells from healthy controls (HCs) and in patients with active ulcerative colitis (aUC), quiescent ulcerative colitis (qUC), active Crohn’s disease (aCD) and quiescent Crohn’s disease (qCD) were allowed to migrate towards gut-homing chemo-attractants including CCL2, CCL25 and MadCam1, following conditioning with anti-TNF drugs (black bars) or the anti-α4β7 drug (grey bars) and compared with non-conditioned cells (white bars). Migrated dendritic cell subsets (DC) including pDCs, cDC2 and cDC1, were identified in Figure 1. All results were relativized with respect to the migration towards a non-supplemented culture medium (basal condition, dotted line) (mean ± SEM *n* = 13–15). Two-way ANOVA with the subsequent post hoc comparison was performed in order to determine the migration differences within each subset and condition with respect to the basal or spontaneous migration (displayed as *) and to compare the migration within each patient group between each culture condition (displayed as #). *p*-values < 0.05 were considered significant (* < 0.05, ** < 0.01, *** < 0.001) (# < 0.05, ## < 0.01).

## Data Availability

Data is contained within the article and Appendix A.

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
