# Peer review of "Differential Effects of Anti-TNFα and Anti-α4β7 Drugs on Circulating Dendritic Cells Migratory Capacity in Inflammatory Bowel Disease"

_biomedicines, 2022, doi:10.3390/biomedicines10081885_

Round 1

Reviewer 1 Report

The authors aimed to test the effect of anti-inflammatory drugs in the migration potential of dendritic cells. While the research question is relevant, the study design has some flaws. Questions and suggestions on how to improve the study are outlines below:

The variety of methods is very limited and additional techniques should be employed to confirm the data.

The statistical analysis is unclear and authors did not explain how and whether they tested for normal distribution and did not explain the subsequent statistical approaches. In addition, the indication of significance (asterisks) appears very random. 

The introduction contains a few statements that are not completely wrong but should be more precise. Please check the original references and ensure that the key information is given correctly in this manuscript.

The second and third column in Figure 4 have exact the same labeling non the t-axis.

Author Response

The authors aimed to test the effect of anti-inflammatory drugs in the migration potential of dendritic cells. While the research question is relevant, the study design has some flaws. Questions and suggestions on how to improve the study are outlines below:

The variety of methods is very limited and additional techniques should be employed to confirm the data.

It is true that flow cytometry is the main technique that we have used given our extensive experience on this technique. Nevertheless, flow cytometry characterization of stimulated cells has been largely used in the literature, not just to phenotype the cells but also to assess the migration capacity as published in top-tier journals (e.g., Bernardo et al CMGH 2016, Mann et al Gut 2016, Bernardo et al Mucosal Immunol 2018, etc.), Therefore, we consider that although we have certainly used one main technique, it is more than valid to address the issues discussed in this research manuscript in our humble opinion.

The statistical analysis is unclear and authors did not explain how and whether they tested for normal distribution and did not explain the subsequent statistical approaches. In addition, the indication of significance (asterisks) appears very random.

The reviewer is right when says that normality should be testes. A Shapiro-Wilk normality test was used and the normality can be assumed in the most of cases. According to the data that not follow a normal distribution, systematical review reported that in terms of type I error ANOVA test is robust even in cases of non-normality distributions (Blanca JA et al 20017). We include the follow paragraph in the material and method section:

Line 150: Data were analyzed using GraphPad Prism 6.01 software (San Diego, CA, USA) by Shapiro -Wilk normality test. In the most of the cases the normality was accepted, but following the indication of previous systematic review[8] the analysis was continued with parametric test. In all cases one-way or two-way analysis of variance (ANOVA) (with or without repeated measures) was carried out. Subsequent Tukey post-hoc correction was applied as detailed in each figure legend. The significance threshold was fixed at p < 0.05 in all cases.

The introduction contains a few statements that are not completely wrong but should be more precise. Please check the original references and ensure that the key information is given correctly in this manuscript.

Line 67: Although these compounds usually modulate the tissue cytokine milieu and therefore the phenotype and function of circulant and tissue resident immune cells

Line 83: Is well know, that alteration in human intestinal DC during UC enhance a Th2 immune response with the consequence loss of cytokine involve in the epithelial barrier maintenance

Line 89: Hence, although GI mucosa DC phenotype differ in IBD patients and controls [5,6], and in the case of CD the DC homing markers correlates with the phenotype of the disease [7], the mechanisms governing these migrations and whether biological drugs can modulate them are currently unknown. Therefore, this study aimed to analyze cDC and pDC migratory capacity towards the GI mucosa, and whether biological drugs can modulate this mechanism in health and IBD.

The second and third column in Figure 4 have exact the same labeling non the t-axis.

Please accept our apologies. We have now changed the labeling on the axis in the figure 4

Reviewer 2 Report

I read the manuscript "Differential effects of anti-TNFα and anti-α4β7 drugs on circulating dendritic cells migratory capacity in inflammatory bowel disease" with great interest. The immunomodulatory targets and new diagnostic markers in IBD are of considerable interest, and it is a very well-written and scientifically sound manuscript. Therefore, I have only a few minor comments: 

- Introduction: I believe some references are needed: for example, line 43 after the definition of IBD. I can see that the authors refer the reference one, but it is not the proper reference for the definition. Other places have the same issue. So I recommend authors go through the introduction and use the appropriate references correctly.

- Materials and methods: please also provide the small details in the methodology to make it replicable for the readers. 

- Results: I wonder why the researchers preferred to use SEM instead of SD. Since they mean, actually quite different things and SEM is primarily recommended in larger sample sizes, mostly over 200.

- Discussion: Please discuss the limitations and strengths of the study.

Author Response

I read the manuscript "Differential effects of anti-TNFα and anti-α4β7 drugs on circulating dendritic cells migratory capacity in inflammatory bowel disease" with great interest. The immunomodulatory targets and new diagnostic markers in IBD are of considerable interest, and it is a very well-written and scientifically sound manuscript. Therefore, I have only a few minor comments:

- Introduction: I believe some references are needed: for example, line 43 after the definition of IBD. I can see that the authors refer the reference one, but it is not the proper reference for the definition. Other places have the same issue. So I recommend authors go through the introduction and use the appropriate references correctly.

We thank to the reviewer for his comments. We included new references along the introduction as suggested. The references included are detailed below:

Line 43: [1]

Line 52 [2]

Line 53 [3]

Line 78 [4,5]

- Materials and methods: please also provide the small details in the methodology to make it replicable for the readers. 

We have included the following changes in materials and methods:

Line 114: In all cases, a Live/Dead fixable near-IR dead cell stain kit (Molecular Probes) was added to the cells during 1 min at room temperature prior to performing antibody staining, hence allowing the exclusion of dead cells from the analysis

Line 123: PBMC from HC and IBD patients were also cultured in complete medium (1 million PBMC/1 ml) in a 24 well cell culture plate at 37 ºC for 18 h

Line 133: Briefly, cells from different groups with or without conditioning were seeded in the apical chamber at 200.000 PBMC per well. Then in the basal chamber 200µl of medium with the corresponding chemoattractant were added. After 4h, migrated cells in the basal chamber were further harvested and analyzed by flow cytometry.

- Results: I wonder why the researchers preferred to use SEM instead of SD. Since they mean, actually quite different things and SEM is primarily recommended in larger sample sizes, mostly over 200.

SEM is calculated by taking the standard deviation and dividing it by the square root of the sample size. Taking into account that the sample size is not the same in all groups we decide to use SEM because this parameter consider the sample size. Having said that, of course we would be delighted to change SEM into SD if the reviewer considers that this were a must.

- Discussion: Please discuss the limitations and strengths of the study.

We have now included in the discussion the follow paragraph:

Line 332: However, the results derived from the current work should be validated by in vivo studies to further understand the biological drugs mechanism of action and for elucidate future diagnostic and prognostic biomarkers than could be implement to the clinical practice.

Reviewer 3 Report

In this manuscript, the authors presented data on "cell homing" surface markers and transwell chemotaxis experiments in circulating dendritic cell subsets from patients with inflammatory bowel disease (IBD).

The main findings include: 1. CCR9 expression in cDC2 is increased in patients from quiescent IBD; 2. chemotaxis towards CCL2 but not CCL25 or MacCam1 is increased in pDC & cDC2; 3. increased chemotaxis to CCL2 in anti-TNF alpha treated cDC2. The authors speculated that cDC2 migration to intestinal mucosa may play a role in IBD, and has the potential be be a diagnostic biomarker.

The research is clearly designed with adequate case numbers in each group. However the authors overinterpreted the data to reach conclusions. There was a lack of hypothesis testing process in this manuscript and therefore no meaningful conclusion can be derived. The lack of consistent effects of anti-a4b7 in DC chemotaxis towards its ligand MadCam1 challenges the relevance of chemotaxis phenotype to disease activity. It is a pity that no further experiments/analyses were designed to explore the mechanitic correlation of CCR9 upregulation and lowered disease activity in IBD. 

Specific comments

1. The manuscript is written in adequate language.

2. There is a lack of introduction to pathophysiology mechanism of DC cell homing to IBD disease severity.

3. Transwell experiments are not adequate to model DC homing to intestinal mucosa.

4. The statistics appear inadequate as multiple comparisons were made but significance level was only set to p=0.05

5. The authors need to specify how to obtain DC subpopulations for transwell experiments. Are these cells sorted to perform chemotaxis tests? Where are the positive control results for the procedure?

Author Response

In this manuscript, the authors presented data on "cell homing" surface markers and transwell chemotaxis experiments in circulating dendritic cell subsets from patients with inflammatory bowel disease (IBD).

The main findings include: 1. CCR9 expression in cDC2 is increased in patients from quiescent IBD; 2. chemotaxis towards CCL2 but not CCL25 or MacCam1 is increased in pDC & cDC2; 3. increased chemotaxis to CCL2 in anti-TNF alpha treated cDC2. The authors speculated that cDC2 migration to intestinal mucosa may play a role in IBD, and has the potential be be a diagnostic biomarker.

The research is clearly designed with adequate case numbers in each group. However the authors overinterpreted the data to reach conclusions. There was a lack of hypothesis testing process in this manuscript and therefore no meaningful conclusion can be derived. The lack of consistent effects of anti-a4b7 in DC chemotaxis towards its ligand MadCam1 challenges the relevance of chemotaxis phenotype to disease activity. It is a pity that no further experiments/analyses were designed to explore the mechanistic correlation of CCR9 upregulation and lowered disease activity in IBD.

We thank to the reviewer for its comments. It is true that in the current work MadCam1 did not exert a clear effect over the DC migration. Nevertheless, in our model we specifically focused in cDC rather that T-cells so we do not consider that our results challenge the relevance of chemotaxis as anti-a4b7 may be eliciting its beneficial effects in IBD over T-cells (or any other immune subset) rather than over cDC. Besides, and although pro-inflammatory macrophages are expanded in the IBD mucosa, monocytes hardly express any a4b7 expression (data not shown) and infiltrate the mucosa in a CCR2 dependent manner. Therefore, we just consider that our findings add further insight into the mechanism of action of the different biologics by highlight that these compounds may elicit different effects over different immune subsets. Given the relevance of the reviewer comment, we would be delighted to provide further information about this issue in the discussion if the reviewer finds it relevant.

Regarding the CCR9 results, we agree with the reviewer about the relevance of the findings. Nevertheless, we could not follow this path in this manuscript since, when we found the results, all patients had been recruited. Having said that, this is a new line of research in our lab given which we would be delighted to comment about if the reviewer finds it relevant.

Specific comments

  1. The manuscript is written in adequate language.
  2. There is a lack of introduction to pathophysiology mechanism of DC cell homing to IBD disease severity.

We included the following changes in the introduction

Line 83: Alterations in human intestinal DC during UC enhance a Th2 immune rendering a loss of cytokine production involved in the epithelial barrier maintenance[9]. Indeed, cDC are thought to migrate to the GI tissue by the expression of the surface integrin α4β7, which interacts with the Mucosal Vascular Addressing Cell Adhesion Molecule 1 (MadCam1) expressed by vascular endothelial cells; besides they also migrate by CCR9 binding to its ligand (CCL25) expressed by small bowel epithelial cells as well as by CCR2 [6]. Therefore, and although cDC migration towards the GI mucosa is enhanced in IBD [10,11] (with cDC homing marker expression correlating with the phenotype of the disease[12]), the mechanisms underlying these migrations and whether biological drugs can modulate them are currently unknown. Therefore, this study aimed to analyze cDC and pDC migratory capacity towards the GI mucosa, and whether biological drugs can modulate this mechanism in health and IBD.

  1. Transwell experiments are not adequate to model DC homing to intestinal mucosa.

We nevertheless disagree with the reviewer as transwell experiments are widely used to evaluate the migratory capacities of different cell types in different pathologies (Velecela V et al 2016; Peng Yang et al 2020; Ponath P et al 2000). If the reviewer consider that this would be important, then of course we would be delighted to further explain the experiments in the manuscript.

  1. The statistics appear inadequate as multiple comparisons were made but significance level was only set to p=0.05

We agree with the reviewer on the issue and hence why ad-hoc correction for multiple comparisons were applied and all analyses. We apologize that we had not explained that in material and methods as we do know.

  1. The authors need to specify how to obtain DC subpopulations for transwell experiments. Are these cells sorted to perform chemotaxis tests? Where are the positive control results for the procedure?

We did not isolate DC from patients as we wanted to have a more “physiological model” by allowing them to migrate within the PBMC setting. Also, and as stated to reviewer #4, prior dosimetry experiments were performed while model optimization so we would be delighted to provide further information on the topic if the reviewer considers that this is a must.

Reviewer 4 Report

The MS describes an interesting and novel study investigating the mechanisms of action of biological IBD treatments. In this study, the expression of several homing markers and the migratory profile of circulating DC subsets towards intestinal chemo-attractants were evaluated, and the effect of biological drugs (anti-TNF alpha or anti integrin alpha4 beta7) were evaluated in healthy vs IBD patients. Overall, the authors propose that a key role for cDC2 migration towards the intestinal mucosa in IBD. The experiments are carefully performed, and the study has pathophysiological relevance to IBD. The manuscript is very well written, and the data are well presented.   However, the paper could be strengthened by several specific revisions, as detailed below:  

The experiments have been performed at only one dose of infliximab and vedolizumab: the authors should justify how they determined these unique doses. In addition, it would be important for the authors to discuss and propose the new immunomodulatory targets to treat IBD.  Are these targets would be different in ulcerative colitis and Crohn’s disease? How the authors envisage to deliver specifically drugs to these targets? Finally, it would be important to validate these findings by taking some in vivo approaches. At least the authors should discuss the possible limitations of the in vitro approaches they have taken for the presented experiments.

Author Response

The MS describes an interesting and novel study investigating the mechanisms of action of biological IBD treatments. In this study, the expression of several homing markers and the migratory profile of circulating DC subsets towards intestinal chemo-attractants were evaluated, and the effect of biological drugs (anti-TNF alpha or anti integrin alpha4 beta7) were evaluated in healthy vs IBD patients. Overall, the authors propose that a key role for cDC2 migration towards the intestinal mucosa in IBD. The experiments are carefully performed, and the study has pathophysiological relevance to IBD. The manuscript is very well written, and the data are well presented.   However, the paper could be strengthened by several specific revisions, as detailed below:

The experiments have been performed at only one dose of infliximab and vedolizumab: the authors should justify how they determined these unique doses.

We thank the reviewer for its accurate comments. Titration experiments were carried out using doses from 1 to 1000µg/ml for each drug. The minimum dose what shown effect was selected for continue with the rest of experiments. If the reviewer consider that this would be important, then of course we would be delighted to further justify our decision in the manuscript.

In addition, it would be important for the authors to discuss and propose the new immunomodulatory targets to treat IBD. Are these targets would be different in ulcerative colitis and Crohn’s disease? How the authors envisage to deliver specifically drugs to these targets?

We have included the next paragraph in the main text

Line 292: In addition, CCR2 has been stipulated as a homing cDC target in the colonic mucosa[6]. In this regard, we hereby have observed that CCL2 (one of CCR2 ligands) exerts a high cDC chemoattractant effect. Therefore, CCR2 could be evaluated as a new immunomodulatory target for UC and colonic CD. Besides, some in vivo works demonstrate beneficial effect in experimental colitis when the number of CCR2+ cells is reduced in the inflamed colon and peripheral blood [7][27].

Finally, it would be important to validate these findings by taking some in vivo approaches. At least the authors should discuss the possible limitations of the in vitro approaches they have taken for the presented experiments.

We have included in the discussion the follow paragraph:

Line 332: However, the results derived from the current work should be validated by in vivo studies to further understand the biological drugs mechanism of action and for elucidate future diagnostic and prognostic biomarkers than could be implement to the clinical practice.

Round 2

Reviewer 3 Report

I have read the revisions as well as responses made by the authors. The changes made to the introduction and to the statistic methods are reasonably acceptable.

In general, the major concerns about a lack of mechanism investigation and results overinterpretation persist. Chemotaxis in transwell without tissue specific context is a very generalized experiment and is not suitable to be the only evidence to investigate DC migration into intestinal mucosa. It is also questionable to be the only data to claim for modulatory role of DC tropism on IBD activity, especially when the data is not consistent with findings from surface marker analysis.

It is difficult to imagine how to enumerate subpopulation DC that has migrated in transwells if they have not been sorted in the very first beginning. The only possible approach is to sort the migrated cells. The procedure would suffer from small number of starting cells and the results are expected to be very variable. Therefore quality control for this assay should be provided if I am to interpret this data.

Author Response

I have read the revisions as well as responses made by the authors. The changes made to the introduction and to the statistic methods are reasonably acceptable.

In general, the major concerns about a lack of mechanism investigation and results overinterpretation persist. Chemotaxis in transwell without tissue specific context is a very generalized experiment and is not suitable to be the only evidence to investigate DC migration into intestinal mucosa. It is also questionable to be the only data to claim for modulatory role of DC tropism on IBD activity, especially when the data is not consistent with findings from surface marker analysis.

We thank to the reviewer for his comments. In the experiments carried out in this work, previous published gut-homing chemoattractants were used (10.1097/SHK.0b013e31823cbff1;10.1196/annals.1326.036;10.1080/14712598.2019.1576631).Of course more experiments are needed to clarify the migratory mechanisms of intestinal DC but we consider that the experiments had tissue context.

In addition, previous articles reported the modulatory role of these cells in the context of IBD. Despite these works and given the complexity of the disease how DCs migrate and the role of biological drugs in this mechanism is still unknown. This was one the reason to stablish this research line. (10.3748/wjg.v17.i33.3761)

It is difficult to imagine how to enumerate subpopulation DC that has migrated in transwells if they have not been sorted in the very first beginning. The only possible approach is to sort the migrated cells. The procedure would suffer from small number of starting cells and the results are expected to be very variable. Therefore quality control for this assay should be provided if I am to interpret this data.

We did not isolate DC from because we wanted to have a “physiological model” and because as the reviewer point out very well given the small proportion of these cell in the total amount of PBMCs the develop of the experiment will be really complicated.

The experiments were carried out by duplicate in all cases and as internal control we used PBMCs without any treatment migrating toward culture medium. We use this basal condition to evaluate the migration of the different DC subsets toward the different chemoattractant.

The number of the different DC subset migrated toward culture medium (basal)CCL2, CCL25 and MadCam1 have being included in the article as supplementary figure 2.